# The Impact of COPD on Hospitalized Patients with Diabetes: A Propensity Score Matched Analysis on Discharge Records

**DOI:** 10.3390/healthcare10050885

**Published:** 2022-05-11

**Authors:** Giuseppe Di Martino, Pamela Di Giovanni, Fabrizio Cedrone, D’Addezio Michela, Francesca Meo, Piera Scampoli, Ferdinando Romano, Tommaso Staniscia

**Affiliations:** 1Department of Medicine and Ageing Sciences, “G. d’Annunzio” University of Chieti-Pescara, 65100 Pescara, Italy; tommaso.staniscia@unich.it; 2Unit of Hygiene, Epidemiology and Public Health, Local Health Authority of Pescara, 65100 Pescara, Italy; 3Department of Pharmacy, “G. d’Annunzio” University of Chieti-Pescara, 65100 Pescara, Italy; pamela.digiovanni@unich.it; 4School of Hygiene and Preventive Medicine, “G. d’Annunzio” University of Chieti-Pescara, 65100 Pescara, Italy; cedronefab@gmail.com (F.C.); michela.daddezio@gmail.com (D.M.); francesca.meo10@gmail.com (F.M.); piera.scampoli@gmail.com (P.S.); 5Department of Infectious Diseases and Public Health, “La Sapienza” University of Rome, 00138 Rome, Italy; ferdinando.romano@uniroma1.it

**Keywords:** diabetes, COPD, propensity score, hospitalization, mortality

## Abstract

(1) Background: Type 2 diabetes is a common comorbidity of chronic obstructive pulmonary disease. Despite the lack of knowledge of the pathophysiological link between diabetes and chronic obstructive pulmonary disease, the presence of diabetes among those with chronic obstructive pulmonary disease is associated with worse outcomes, such as mortality and hospitalization. The aim of this study was to evaluate the impact of chronic obstructive pulmonary disease on in-hospital mortality and prolonged length of stay (PLOS) among patients with diabetes. (2) Methods: The study considered all hospital admissions of patients with diabetes aged over 65 years performed from 2006 to 2015 in Abruzzo, Italy. To compare outcomes between patients with and without chronic obstructive pulmonary disease, a propensity score matching procedure was performed. (3) Results: During the study period, 140,556 admissions of patients with diabetes were performed. After matching, 18,379 patients with chronic obstructive pulmonary disease and 18,379 controls were included in the analyses. Logistic regression analyses showed as chronic obstructive pulmonary disease was associated with in-hospital mortality (OR: 1.10; *p* = 0.036) and PLOS (OR: 1.18; *p* = 0.002). (4) Conclusions: In a cohort of Italian patients, diabetic patients with chronic obstructive pulmonary disease were associated with in-hospital mortality and PLOS. The definition of the causes of these differences aims to implement public health surveillance and policies.

## 1. Introduction

Type 2 diabetes mellitus (T2DM) is a systemic disorder characterized by a chronic hyperglycemic state associated with chronic inflammation and oxidative stress. This leads to micro- and macrovascular damage to many organs, especially the kidney, retina, and cardiovascular system [1,2]. In addition, other factors such as increased obesity, reduced physical activity, increased smoke exposure, corticosteroid consumption, and disease-related inflammation state are also factors associated with T2DM [1,3]. T2DM is a common comorbidity of chronic obstructive pulmonary disease (COPD). COPD is a preventable and treatable disease characterized by chronic respiratory symptoms and airflow limitation due to airway abnormalities [1]. Previous studies suggest that T2DM is much more common in patients with COPD than in non-COPD patients [3,4,5], estimating a high prevalence of diabetes among COPD patients, about 10.1 to 23.0% [6,7]. Although COPD and T2DM represent distinct entities, there might indeed be a pathophysiological link that connects these frequent chronic conditions [8]. In fact, the frequent coexistence of these diseases in the same adult individual supports the hypothesis of common processes sharing their pathogeneses within the same patient [1]. The risk of diabetes in COPD patients seems to be higher in more severe phenotypes (levels 3–4 according to the Global Initiative for GOLD guideline) [4]. This risk was independent of BMI, smoking, and other confounding factors. Despite the lack of knowledge of the pathophysiological link between T2DM and COPD, the presence of diabetes among those with COPD has been shown to be associated with worse outcomes, such as mortality and hospitalization [1,4].

Few studies have investigated the impact of T2DM and COPD on hospital outcomes and few of them were performed in the Italian setting. The aim of this study was to evaluate the impact of COPD on in-hospital mortality and prolonged length of stay among hospitalized patients with diabetes in a large Italian cohort.

## 2. Materials and Methods

### 2.1. Study Design

The study considered all hospital admissions of diabetic patients aged over 65 years performed between January 2006 and December 2015 in the Abruzzo Region, Italy. Abruzzo is a southern Italy region, with about 1.2 million inhabitants and 29 hospitals, divided between 18 public hospitals and 11 private clinics. Abruzzo is one of the Italian regions with the highest prevalence of diabetes (7.4% of the entire population) [9]. Data were collected from hospital discharge records (HDR), which included information on the patient’s demographics, a Diagnosis-Related Group code (DRG) used to classify the admission, and up to six diagnoses and six procedures performed during the hospitalization, coded as per the International Classification of Disease, 9th Revision, Clinical Modification (ICD-9-CM). Only patients with code 250.xx (diabetes diagnosis) as diagnostic code were included. Patients with multiple admissions were counted only for the first event that occurred. Admissions with a duration over 15 days were considered a ‘prolonged length of stay’ (PLOS). This value represented the upper quartile of the length of stay distribution of the study population. Among diabetic patients included all patients with codes 491 (chronic bronchitis), 492 (emphysema), 494 (bronchiectasis), or 496 (chronic airway obstruction, not elsewhere classified), were considered as affected by COPD, as specified by Quan et al. [10].

### 2.2. Statistical Analysis

Quantitative variables were presented as mean and standard deviation (SD). Qualitative variables were presented as frequency and percentage. Basic statistic models are not adequate for this kind of data. In particular, the large cohort considered and the possible presence of confounding factors can lead to biased results. In order to allow an unbiased comparison between patients with and without COPD, a propensity score matching (PSM) was performed to compare outcomes between patients with and without COPD [11]. For PSM, a logistic regression model including age, gender, and all Charlson comorbidity index (CCI) diseases as covariates was used. A 1:1 greedy matching algorithm was used to identify a unique matched control for COPD patients according to the propensity score. All baseline variables included in the matching model are listed in Table 1. The adequacy of covariate balance in the matched sample was evaluated via standardized mean differences between the two groups, with differences of less than 10% indicating a good balance [12]. Unmatched patients were discarded from the analysis. Odds ratios and their 95% confidence intervals (95% CI) for study outcomes were performed using logistic regression models, using study outcomes as a dependent variable and COPD as an independent variable, adjusting for propensity score. ***p***-values less than 0.05 were considered significant. The statistical analysis was performed using IBM SPSS Statistics v23.0 (SPSS Inc., Chicago, IL, USA).

## 3. Results

During the study period, a total of 140,556 admissions of diabetic patients were performed: 18,379 with COPD and 122,177 without COPD. Compared to the unmatched diabetic sample, COPD patients were mainly male and they were more frequently affected by chronic heart failure (CHF) and kidney disease (KD). After the matching procedure, 36,758 patients were included in the analyses: 18,379 with COPD and 18,379 controls, as shown in Figure 1.

After matching, all baseline characteristics showed good balance, with a standardized mean difference of less than 10% for all considered variables, as shown in Table 1.

Congestive heart failure was the most frequent cause of admission in both study groups, causing 3103 (16.9%) admissions among patients with COPD and 2978 (16.2%) among patients without COPD. COPD patients were frequently admitted for respiratory signs (759, 4.1%) but both groups frequently suffered from pulmonary edema and respiratory failure, 2784 (15.1%) among patients with COPD and 474 (2.6%) among patients without COPD as reported in Table 2.

Logistic regression analyses showed as COPD was associated to in-hospital mortality (OR: 1.10; 95% CI 1.01–1.20; *p* = 0.036) and prolonged length of stay (OR: 1.18; 95% CI 1.06–1.31; *p* = 0.002), as reported in Table 3.

A supplementary analysis by gender showed both females (OR: 1.23; 95% CI 1.07–1.43; *p* = 0.005) and males (OR: 1.10; 95% CI 1.00–1.29; *p* = 0.047) were associated with in-hospital mortality. Both genders were also associated with PLOS, as reported in Appendix A.

## 4. Discussion

This study evaluated the association between COPD and hospital outcomes in a large cohort of Italian diabetic patients. The study was conducted during a ten-year period in an Italian region with a high prevalence of diabetes [13,14]. In this study, diabetic patients with COPD were more frequently male and suffered more frequently from CHF and KD. This is in line with previous literature [5,15]. In particular, it is known that the presence of COPD increased the prevalence of heart failure [5,16]. The mechanisms by which the pulmonary changes induced by COPD can lead to these cardiovascular events are not clear. A potential factor is increased arterial stiffness, which is related to chronic systemic inflammation [17]. Systemic markers such as CRP, TNF-α, and IL-6 play an important role in both the progression of COPD and the development of insulin resistance. Subcutaneous and intra-abdominal adipose tissue correlated with circulating levels of IL-6 and TNF-α. The mechanisms underlying lung volume restriction could include a combination of abdominal obesity and reduced pulmonary elasticity caused by cytokine relapses [18]. In addition, it should be highlighted that inflammation is a highly significant risk factor for both morbidity and mortality in elderly people, as demonstrated by previous studies [19,20].

On the other hand, CHF is more prevalent among patients with diabetes compared to the general population [21], so COPD can amplify the impact of diabetes on cardiovascular disease and related outcomes. After the matching procedure, patients with COPD were more likely associated with in-hospital mortality and prolonged length of stay. Few studies have investigated these issues. The study by Parappil et al. [7] showed comorbid diabetes was associated with death and length of stay in patients admitted for exacerbation of COPD. Probably the hyperglycemic state of patients with diabetes impairs the response to COPD treatment [21,22,23]. In addition, it is plausible that COPD patients with diabetes admitted for infectious diseases respond less well to antibiotic treatment [7]. Probably the main point contributing to the increased risk of death was COPD exacerbation: diabetes is a strong predictor of COPD exacerbation and they both contribute to worse outcomes [24,25]. A prolonged LOS can be explained by the need to get comorbid conditions under control, and the presence of COPD can, among patients with diabetes, cause an additional need for care [26]. These results are in line with previous studies performed in Italy, where comorbidities such as diabetes were associated with an increase in mortality and LOS [27,28]. The prolonged length of stay can be also explained by the possible exacerbation of the hyperglycemic state that occurred after steroid treatment which requires more time to be controlled among COPD patients [7]. Since it correlates with a longer hospital stay and a higher risk of infection, hyperglycemia during hospital admission has been associated with a bad prognosis [29]. For this reason, in patients with COPD and type 2 diabetes, physicians should consider treatment interactions associated with the coexistence of these conditions [8]. It is ascertained that the hyperglycemic state could have adverse effects in acute illness, particularly among patients with COPD [8,30]. This study also confirmed that HF was one main cause of admission both for COPD+ and COPD- diabetic patients. Considering that one-third of COPD patients have been affected by CHF, the contemporary exacerbation of both COPD and CHF will cause a prolonged duration of admission [31]. Supplementary analysis showed that gender is an important risk factor for mortality and LOS. These results showed as both genders were significantly associated with in-hospital mortality. The coexistence of both diabetes and COPD seems to have a strong impact on females compared to males and this point should be deeply investigated in future studies. In previous studies on comorbidities, the male gender reported a higher in-hospital mortality rate compared to women [32].

### 4.1. Strengths and Limitations

The major strength of this study is represented by the large cohort evaluated, making the study results very robust. Propensity score matching is also a robust methodology to limit possible confounders, as shown in previous studies on diabetes [33,34,35]. In fact, despite COPD being a male-oriented disease, the use of PSM allows to correct the influence of comorbidities and patient characteristics such as gender. In parallel, some limitations should be considered. Firstly, the identification of diagnosis is based on ICD-9-CM codes that cannot consider the severity of each condition. Secondly, the use of administrative data may be limited by the reliability of some information such as drug therapy and other clinical information. In particular, performance status cannot be measured but some diseases included in CCI, such as dementia, rheumatological diseases, and hemi/paraplegia, are important contributors to poor performance status in older people. Finally, the real prevalence of some comorbidity could be underestimated, due to underreporting in the hospital discharge registry.

### 4.2. Unanswered Questions and Future Research

This study can be used as the starting point of future research about the interaction between diabetes and COPD during hospitalization. In particular, it will be useful to develop a prediction model estimating the added number of hospitalization days caused by COPD among patients with diabetes. In addition, factors associated with a PLOS in diabetic patients with COPD can help in developing tailored treatment.

## 5. Conclusions

In a large cohort of Italian diabetic patients, COPD was associated with in-hospital mortality and prolonged length of stay, compared to patients without COPD. The definition of the causes of these differences aims to improve surveillance systems and implement better policies. The understanding of whether treatment of COPD influences the course of diabetes mellitus or if it is altered by the presence of the concomitant comorbid disease. It is also important to know whether the treatment of type 2 diabetes can alter the natural history of concomitant COPD.

## Figures and Tables

**Figure 1 healthcare-10-00885-f001:**
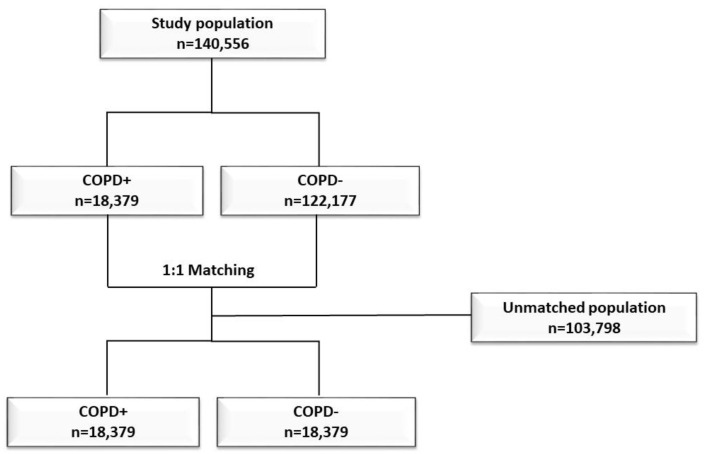
Flow-chart of the study population.

**Table 1 healthcare-10-00885-t001:** Baseline characteristics of matched and unmatched populations.

	COPD+ (*n* = 18,379)	Unmatched COPD− (*n* = 122,177)	Matched COPD− (*n* = 18,379)	Standardized Mean Difference
Demographic				
Age *mean ± SD*	78.5 ± 6.8	77.5 ± 7.1	78.5 ± 7.1	0.003
Women *n(%)*	6420(34.9)	63,125(51.7)	6569(35.7)	−0.017
Medical Hystory *n*(%)				
Ischemic heart disease	871(4.7)	8340(6.8)	891(4.8)	−0.005
Chronic heart failure	5118(27.8)	20,448(16.7)	5038(27.4)	0.010
Peripheral vascular disease	943(5.1)	8594(7.0)	946(0.4)	−0.001
Cerebrovascular disease	1976(10.8)	23,046(18.9)	1930(10.5)	0.008
Dementia	313(1.7)	3050(2.5)	307(1.7)	0.003
Peptic ulcer	53(0.3)	818(0.7)	55(0.3)	−0.002
Cancer	1056(5.7)	9508(7.8)	1038(5.6)	0.004
Metastatic cancer	146(0.8)	2576(2.1)	148(0.8)	−0.001
Reumatologic disease	128(0.7)	1133(0.9)	112(0.6)	0.010
Hemiplegia or paraplegia	45(0.2)	768(0.6)	49(0.3)	−0.004
Liver disease	62(0.3)	1224(1.0)	68(0.4)	−0.006
Kidney disease	2009(10.9)	13,111(10.7)	2021(11.0)	−0.002
AIDS/HIV	11(0.1)	57(0.1)	11(0.1)	0.002

**Table 2 healthcare-10-00885-t002:** Principal diagnosis of hospitalization diabetic patients with and without COPD.

COPD+ (*n* = 18,379)	*n*%	COPD− (*n* = 18,379)	*n*%
Congestive heart failure	3103(16.9)	Congestive heart failure	2978(16.2)
Pulmonary edema and respiratory failure	2784(15.1)	Diabetes	524(2.9)
Respiratory signs, symptoms and minor diagnoses	759(4.1)	Pulmonary edema and respiratory failure	474(2.6)
Bronchiolitis and rsv pneumonia	395(2.1)	Other circulatory system diagnoses	469(2.6)
Others	11,338(61.8)	Others	13,934(75.8)

**Table 3 healthcare-10-00885-t003:** Outcomes comparison between diabetic patients with and without COPD.

Outcomes	COPD+ (*n*%)	COPD− (*n*%)	Odds Ratio *	95% CI	*p*-Value
In-hospital mortality	1056(5.7)	968(5.3)	1.10	1.01–1.20	0.036
Lenght of stay over 15 days	5074(27.6)	4595(25.0)	1.18	1.06–1.31	0.002

* All models were adjusted for propensity score.

## Data Availability

Data available on request due to privacy and ethical restrictions.

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
