# Peer review of "The Impact of COPD on Hospitalized Patients with Diabetes: A Propensity Score Matched Analysis on Discharge Records"

_healthcare, 2022, doi:10.3390/healthcare10050885_

Round 1

Reviewer 1 Report

Accept

Author Response

We thanks the referee for the comments.

Reviewer 2 Report

The aim of this study was to assess the impact of COPD on hospital mortality and prolonged stay in hospitalized patients with diabetes in a large Italian cohort.

This work confirms that the presence of diabetes in patients with COPD is associated with more mortality and hospitalization. Logistic regression models are used to validate this relationship.

This result coincides with what was expected and there is no surprise. That is why this work shows a result that is not spectacular. Perhaps these results are a good starting point to consider other types of predictions.

The authors could explain other research projects that could be derived from this article. Maybe predict the number of days of hospitalization, etc...

Author Response

  • The authors could explain other research projects that could be derived from this article. Maybe predict the number of days of hospitalization, etc...

Reply: We thanks the referee for the comments. We added a specific section in discussion chapter (see Section 4.2 - Unanswered questions and future research);

Reviewer 3 Report

In this paper, Di Martino and colleagues strived to interconnect chronic obstructive pulmonary disease with type 2 diabetes in hospitalized patients in the Italian region of Abruzzo.

The study includes a solid number of subjects and the methodology for the data analysis is clearly defined and well-described.

I do have some small comments:

  • although not many studies have addressed this issue, the Discussion section should be extended. perhaps the authors could ellaborate further on the causes and mechanisms interconnecting COPD, DM2 and PLOS, since this was an important aspect interconnecting both diagnoses.
  • in line 75, the patients were considered as affected by COPD, as specified by what?
  • I would recommend the authors to carefully check the manuscript for typos, and to chcek for a proper formatting of the list of references as per the instructions for authors.

Author Response

  • although not many studies have addressed this issue, the Discussion section should be extended. perhaps the authors could elaborate further on the causes and mechanisms interconnecting COPD, DM2 and PLOS, since this was an important aspect interconnecting both diagnoses.

Reply: We thanks the referee for the comments. We added two more sentences in discussion section focusing on this point (references23 and 24);

  • in line 75, the patients were considered as affected by COPD, as specified by what?

Reply: We thanks the referee for the comments. Sorry for the mystake. We added the proper reference. (Ref 10 - Quan H, Sundararajan V, Halfon P, Fong A, Burnand B, Luthi JC, Saunders LD, Beck CA, Feasby TE, Ghali WA. Coding algorithms for defining comorbidities in ICD-9-CM and ICD-10 administrative data. Med Care. 2005 Nov;43(11):1130-9. doi: 10.1097/01.mlr.0000182534.19832.83. PMID: 16224307).

  • I would recommend the authors to carefully check the manuscript for typos, and to chcek for a proper formatting of the list of references as per the instructions for authors.

Reply: We thanks the referee for the comments. We revised the manuniscript and corrected all typos.

This manuscript is a resubmission of an earlier submission. The following is a list of the peer review reports and author responses from that submission.

Round 1

Reviewer 1 Report

Research article titles as “The impact of COPD on hospitalized patients with diabetes: a propensity score matched analysis on discharge records” by Giuseppe Di Martino et al. gives an important information about link between COPD and diabetes. Article is well written and describes something novel but following changes are required before its considered for publication:

  1. Introduction section should expand on link between diabetes and COPD. In-depth description of possible mechanism and/or references from current body of knowledge on this topic may be helpful
  2. It is very interesting that COPD affects the mortality rate in diabetic patients but as COPD is a male-oriented disease, it may be interesting to elaborate the findings in COPD females vs COPD males and then compare it with mortality and hospitalization with diabetes. Few lines in results section in this respect will be helpful.
  3. How authors will explain the low mortality rates and PLOS in females with same degree of COPD
  4. Elaborate the codes or give references in this line“Among diabetic patients included, all patients with codes 490, 492, 494 or 496, were considered as affected by COPD”
  5. Correct an abbreviation mistake at line number 31

Reviewer 2 Report

The present submission aims to investigate the effects of COPD-Diabetes comorbidity on in-hospital mortality and length of stay after hospitalization.

Whereas I found the topic interesting, I feel the outcome of the study is somewhat disappointing. Below my suggestions on how to improve the paper:

Abstract: Please do not use abbreviations here already. These can be introduced in the main text.

Section 1:

The Introduction is quite underdeveloped and needs more detail and overall a better motivation and structure and information on the approach the authors take. I list a couple of points the authors need to work on:

  • Line 31: The authors introduce Type 2 diabetes as (T‘‘DM), which is odd, and later say „T2DM“. I think there is just a typo in the brackets.
  • Lines 31-34: Other than the biological effects and health outcomes, a neutral reader will be interested in behavioral and environmental factors associated with Type 2 diabetes. Therefore, please give a short literature review on those.
  • Please also add a short paragraph on COPD. Although most readers will roughly be familiar with it, it seems necessary to me to write a couple of notes on how it‘s induced and what health outcomes are associated with it would be welcome.
  • Lines 44-46: I would start a new paragraph before „The aim…“. Overall, there is no description of how you approach it and a short overview of the structure of the paper, as would be common for a scientific paper. Please add this information.

Section 2.2:

I think the ethical approval should be cast out from the main text. This is legislative information which you can add in the ethical statements after the paper. The reader shouldn’t be bothered with this.

Section 2.3:

  • The methods are only described in a short paragraph, which does not contain details on the variables and just refers verbally to the approach and the software it was performed in, which raises questions whether the authors have themselves thoroughly understood what they have done there and why. In my mind, that's a crucial problem of the paper. If the readers cannot be sure what you have done and whether you know what you have done yourselves and the merits of your methods, how can they trust your results? I also lack a proper explanation on why the authors chose that concrete approach? Is there a reason, other than that the algorithm is nicely programmed in some package (which would not be a sufficiently good reason), that you proceeded the way you did? 
  • I particularly miss the detailed (mathematical) description of the matching. I do not understand the exact model behind this.
  • How exactly has the logistic regression been done, and what have been the exact results? I did not understand this from the tables.

Section 4:

Overall, this section seems fine. My only point here is that I find it critical calling out previous studies as "poor", while not giving plausible justification why you do so and giving convincing evidence what these studies did wrong and what your study is doing better. Again, without giving a detailed and comprehensive description of your approach that's something the reader cannot assess, and it remains a bold, yet not evidenced, statement from your side. 

Section 4.1:

I find this section unconvincing and again too short. I did not find any convincing argument for the study other than a large sample (which is, of course, a good one). The authors did not show a thorough and critical reflection of their own methodological approach, which may reflect a non-sufficient understanding of the methods by the authors.

The paper, in general, reads more like an epidemiological report than a scientific paper. Overall, I see too many flaws in the paper, especially the presentation of and the justification for methodological approaches.